# Optimal Sequencing and Predictive Biomarkers in Patients with Advanced Prostate Cancer

**DOI:** 10.3390/cancers13184522

**Published:** 2021-09-08

**Authors:** Carlo Cattrini, Rodrigo España, Alessia Mennitto, Melissa Bersanelli, Elena Castro, David Olmos, David Lorente, Alessandra Gennari

**Affiliations:** 1Medical Oncology, “Maggiore della Carità” University Hospital, 28100 Novara, Italy; carlo.cattrini@maggioreosp.novara.it (C.C.); alessia.mennitto@maggioreosp.novara.it (A.M.); alessandra.gennari@uniupo.it (A.G.); 2Department of Translational Medicine (DIMET), University of Eastern Piedmont (UPO), 28100 Novara, Italy; 3Department of Internal Medicine and Medical Specialties (DIMI), University of Genoa, 16132 Genoa, Italy; 4Urology Unit, Hospital Regional de Málaga, University of Malaga, 29910 Málaga, Spain; espanarodrigo@outlook.com; 5Medical Oncology Unit, University Hospital of Parma, 43126 Parma, Italy; bersamel@libero.it; 6Department of Medicine and Surgery, University of Parma, 43126 Parma, Italy; 7Genitourinary Cancer Translational Research Group, Instituto de Investigación Biomédica de Málaga, 29010 Málaga, Spain; elena.castro@ibima.eu; 8Medical Oncology, UGCI, Hospitales Universitarios Virgen de la Victoria y Regional de Málaga, 29010 Málaga, Spain; 9Prostate Cancer Clinical Research Unit, Spanish National Cancer Research Centre, 28029 Madrid, Spain; david.olmos@ibima.eu; 10Genitourinary Cancer Translational Research Group, The Institute of Biomedical Research in Málaga, 29010 Málaga, Spain; 11Medical Oncology, Hospital Provincial de Castellón, 12002 Castellón de la Plana, Spain

**Keywords:** metastatic castration-resistant prostate cancer, metastatic hormone-sensitive prostate cancer, metastatic hormone-naïve prostate cancer, nonmetastatic castration-resistant prostate cancer, chemotherapy, androgen-receptor signaling inhibitors, LuPSMA, PARP inhibitors, ipatasertib

## Abstract

**Simple Summary:**

Several strategies have demonstrated the ability to improve the survival of patients with both metastatic and nonmetastatic prostate cancer. The old backbone of androgen-deprivation monotherapy has been disrupted in the hormone-sensitive setting, and several options have been introduced for the management of the castration-resistant disease. However, no optimal sequencing is still defined, and few randomized comparisons are currently available to identify the approach that maximizes the long-term benefit for these patients. This comprehensive review aims at resuming the current evidence on this topic to help physicians during the treatment choice for patients with advanced prostate cancer.

**Abstract:**

The treatment landscape of advanced prostate cancer has completely changed during the last decades. Chemotherapy (docetaxel, cabazitaxel), androgen-receptor signaling inhibitors (ARSi) (abiraterone acetate, enzalutamide), and radium-223 have revolutionized the management of metastatic castration-resistant prostate cancer (mCRPC). Lutetium-177–PSMA-617 is also going to become another treatment option for these patients. In addition, docetaxel, abiraterone acetate, apalutamide, enzalutamide, and radiotherapy to primary tumor have demonstrated the ability to significantly prolong the survival of patients with metastatic hormone-sensitive prostate cancer (mHSPC). Finally, apalutamide, enzalutamide, and darolutamide have recently provided impactful data in patients with nonmetastatic castration-resistant disease (nmCRPC). However, which is the best treatment sequence for patients with advanced prostate cancer? This comprehensive review aims at discussing the available literature data to identify the optimal sequencing approaches in patients with prostate cancer at different disease stages. Our work also highlights the potential impact of predictive biomarkers in treatment sequencing and exploring the role of specific agents (i.e., olaparib, rucaparib, talazoparib, niraparib, and ipatasertib) in biomarker-selected populations of patients with prostate cancer (i.e., those harboring alterations in DNA damage and response genes or *PTEN*).

## 1. Introduction

The treatment landscape of advanced prostate cancer has completely changed in recent years. Several treatment options have demonstrated the ability to improve the overall survival (OS) of patients with metastatic hormone-sensitive prostate cancer (mHSPC) and metastatic castration resistant prostate cancer (mCRPC) when added to the androgen-deprivation therapy (ADT). These treatments include chemotherapy with docetaxel and cabazitaxel, androgen-receptor signaling inhibitors (ARSi), radium-223, and radiotherapy to primary tumor. Phase 3 trials have also shown that the addition of ARSi to ADT improves the outcomes of patients with nonmetastatic castration resistant prostate cancer (nmCRPC). However, no standardized approach to correctly sequence all of these treatment options is still defined. Few randomized trials have compared these different strategies, and the majority of information is provided by retrospective series and secondary analyses. This comprehensive review aims at providing the most convincing evidence on the best sequencing of agents in different settings of advanced prostate cancer and to discuss current data that support the use of specific biomarkers during the treatment choice.

## 2. Optimal Sequencing in mHSPC, nmCRPC and mCRPC

### 2.1. Selection of First-Line Treatment

#### 2.1.1. First-Line mHSPC

Long-term ADT has been the treatment of choice in the mHSPC setting for decades, with an estimated median OS of about 3.5 years in contemporary series [1,2]. In recent years, however, the addition of chemotherapy, ARSi, and radiotherapy to primary tumor to ADT have been shown to provide a significant survival benefit in patients with mHSPC [3].

The addition of docetaxel to ADT demonstrated an OS gain ranging from 10.4 to 16 months in the CHAARTED and STAMPEDE trials, respectively [1,2]. The OS advantage of adding abiraterone acetate to ADT was 16.8 months (53.3 vs. 36.5 months) in the LATITUDE trial [4] and 33.6 months (79.2 vs. 45.6 months) in the STAMPEDE trial [5]. Enzalutamide and apalutamide also provided a significant OS benefit in patients with mHSPC enrolled in the ENZAMET (HR 0.67, 95% CI 0.52–0.86) and TITAN trials (HR 0.67, 95% CI 0.51–0.89) [6,7]. Radiotherapy to the primary tumor prolonged OS in patients with low-volume mHSPC [8]. Overall, the important survival advantage observed in these trials, together with the significant improvement of several secondary endpoints, strongly support the clinical use of these strategies during the first-line treatment of mHSPC (Table 1).

Although these trials have introduced new active options for the treatment of mHSPC, no direct head-to-head comparisons are currently available. Cross-trials comparisons appear to be inappropriate given that the median OS observed in the control arms varies from 36.5 months of the LATITUDE trial to 52.2 months of the TITAN trial. In addition, the majority of retrospective analyses or indirect comparisons are subjected to the bias of different trials’ design, population, treatment duration, and follow-up.

In the only indirect, patient-level comparison available to date, the outcomes from 566 patients treated in the STAMPEDE trial who were contemporarily randomized to abiraterone acetate and docetaxel were assessed. Although abiraterone was associated with longer failure-free and progression-free survival (PFS), there was no difference in terms of OS between abiraterone- and docetaxel-treated patients (HR = 1.16, 95% CI 0.82–1.65) [28].

Clinical variables, such as disease volume or patients’ risk, have been proposed to select the appropriate treatment choice for patients with mHSPC. In the CHAARTED study, a prospective stratification of high- versus low-volume disease (defined as the presence visceral metastases and/or four or more bone lesions, with one or more beyond the pelvis and vertebral bodies) was included in the trial. Docetaxel was found to significantly improve OS in patients with high-volume disease (HR 0.63, 95% CI 0.50–0.79) but not in those with low metastatic burden (HR 1.04, 95% CI, CI: 0.70–1.55, interaction *p* = 0.033) [1]. However, in the STAMPEDE trial, docetaxel added to ADT was found to be superior to ADT alone (HR = 0.81, 95% CI 0.69–0.95), irrespective of metastatic burden (interaction *p* = 0.827) [2]. Similarly, no evidence of heterogeneity of effect between high- and low-volume subgroups was found in the phase 3 trials of enzalutamide and apalutamide [6,7]. The LATITUDE trial of abiraterone acetate was specifically designed to detect a survival benefit in patients with high-risk mHSPC (defined as the presence of at least two high-risk features, including ≥3 bone metastases, visceral metastases, and/or Gleason ≥8). However, no interaction according to disease volume or patient’s risk was found in men treated with abiraterone acetate in the STAMPEDE trial [29]. Furthermore, in the same trial, radiotherapy to primary tumor was found to only improve OS in men with newly diagnosed low-volume mHSPC (HR 0.68, 95% CI 0.52–0.90) but not in those with high-volume disease (HR 1.07, 95% CI 0.90–1.28, interaction *p* = 0.0098) [8]. The interpretation of such subgroup analyses is still matter of debate [30], and the choice of the first-line treatment for mHSPC currently depends on multiple factors, including drugs’ tolerability profile, costs, patients’ characteristics, duration of treatment, and the local reimbursement of specific drugs [3]. The eligibility for chemotherapy is currently an important deciding factor, as many patients show older age, comorbidities, and weak performance status. These patients are not suitable for treatment with docetaxel given its toxicity profile and potentially lethal adverse events (AEs). The use of prednisone in combination with abiraterone acetate can also discourage its initial use in patients with diabetes mellitus. However, to date, no data are available to guide the first-line treatment of mHSPC based on a putative benefit on the subsequent sequencing of agents, and further studies are warranted in this setting. Figure 1 shows the possible sequencing scenarios in patients who are initially treated with ARSi, chemotherapy, and radiotherapy to primary tumors in an mHSPC setting.

#### 2.1.2. First-Line nmCRPC

Patients with nmCRPC show biochemical progression on ADT, with baseline PSA ≥ 2 ng/mL, and no evidence of metastatic disease on conventional imaging (bone scan and computed tomography or magnetic resonance). Before 2018, no standard of care was established for these patients. After initial studies with zoledronic acid in patients with nmCRPC were stopped due to the lack of events [31], subsequent studies were restricted to high-risk patients, defined as those with a PSA doubling time (PSADT) ≤ 10 months. In three randomized studies, apalutamide (SPARTAN trial), enzalutamide (PROSPER trial), and darolutamide (ARAMIS trial) have shown a significant benefit in metastasis-free survival (MFS) and OS over ADT alone, with a good tolerability profile [32,33,34] (Table 1). A recent pooled analysis of patient-level data from these phase 3 trials also supports the use of these agents in men aged 80 years or older, even if they are more likely to experience grade 3 or worse AEs [35].

Subtle differences in trial design, such as the inclusion of patients with lymph node disease in SPARTAN but not in PROSPER or ARAMIS, may limit cross-trial comparisons of efficacy. Although tolerability seemed improved in the ARAMIS trial, where rates of discontinuation due to AEs were not different to placebo (8.9% vs. 8.7%) in contrast to SPARTAN (14.9% vs. 7.3%) or PROSPER (17% vs. 9%) trials, the different reporting of AEs can also limit conclusions on toxicity [36]. Of note, new imaging modalities, such choline positron-emission tomography (PET) or prostate-specific membrane antigen tomography (PET-PSMA), have shown the ability to identify metastatic disease in the majority of non-metastatic patients by conventional imaging [37]. However, metastatic disease by novel imaging techniques was not an inclusion criterion neither in the phase 3 trials including patients with mCRPC nor in those performed in patients with nmCRPC. Therefore, caution should be used when extrapolating the benefit reported in trials where the burden of disease was evaluated with computed tomography or bone scan to patients where mCRPC is defined based on novel imaging modalities. On a methodological basis, treatments approved for nmCRPC should be used for patients with metastatic disease at PET scan and concurrent nonmetastatic disease according to conventional imaging. Although the early use of these ARSi have demonstrated a substantial benefit in terms of OS and they should be offered to patients who meet the criteria of nmCRPC, the impact of these drugs on the subsequent treatment lines remains unclear given the potential emergence of cross-resistance with other ARSi and chemotherapy.

#### 2.1.3. First-Line mCRPC in Patients Pretreated with ADT Monotherapy

Docetaxel (TAX-327 trial), abiraterone acetate (COU-AA-302 trial), and enzalutamide (PREVAIL trial) have all shown a significant survival benefit as first-line therapies for mCRPC and are considered standard options in initial therapy [16,17,38] (Table 1). The current interpretation of these trials is challenging, as enrolled patients had mainly received ADT as prior therapy. However, in the current clinical scenario, the majority of patients have received ARSi or chemotherapy in addition to ADT for mHSPC or nmCRPC. It is not known to what extent the clinical benefit observed in the phase 3 trials of mCRPC would be observed nowadays after treatment with these agents in prior settings. Potential cross-resistance between agents is not fully understood and could significantly limit treatment benefit. The current median OS from first-line therapy is likely lower than that reported in the pivotal COU-AA-302 and PREVAIL trials, since patients are now experiencing longer time in the mHSPC or nmCRPC stages of the disease.

No formal randomized comparison between chemotherapy and ARSi is currently available in the first-line setting of mCRPC. The marked difference in median OS observed in the control arms of the TAX-327 (16.5 months), COU-AA-302 (30.3 months), and PREVAIL (31 months) trials suggests that different patient populations were investigated and cross-trial efficacy comparisons are inappropriate. In a large, real-world, observational study, patients treated with first-line ARSi experienced longer times to progression than those treated with docetaxel, but there was no difference in terms of OS [39]. Additionally, patients with worse baseline prognostic features were more likely to receive first-line docetaxel. Similar results were observed in a sub-analysis of the prospective PROREPAIR-B study [40]. The longer PFS observed in patients treated with ARSi compared to those treated with chemotherapy might be related to the different exposure to treatment, which is continuous with ARSi and limited with docetaxel. Some retrospective data suggest that a short duration of response to prior treatment with ADT predicts a poor response to ARSi [41], whereas docetaxel seems to retain its efficacy in patients experiencing early castration-resistance [42].

Docetaxel remains the first-line taxane of choice in mCRPC based on the results of the FIRSTANA trial, where no difference in survival was observed when comparing first-line docetaxel with cabazitaxel, although cabazitaxel seemed to be better tolerated than docetaxel at the 20 mg/m^2^ dose [43]. Since the trial was designed to demonstrate the superiority in OS (not non-inferiority), cabazitaxel was not approved as a first-line option for mCRPC.

To date, no recommendation based on efficacy can be made for the selection of a first-line ARSi. Indirect comparisons of phase 3 trials have not reported statistically significant differences in OS between abiraterone and enzalutamide both pre- and post-chemotherapy for mCRPC [44], although enzalutamide may better outperform control arms in terms of time to PSA progression, radiographic PFS, and PSA response rate. It must be noted, however, that abiraterone trials (COU-AA-301, COU-AA-302) used prednisone (an agent with known antitumor activity in mCRPC) as a control arm, whereas enzalutamide trials (AFFIRM, PREVAIL) used placebo as a control arm. Results from a recent retrospective study of 3174 patients with chemotherapy-naive mCRPC treated with first-line enzalutamide or abiraterone acetate, those who received enzalutamide had significantly better OS compared to those who were treated with abiraterone (HR 0.84, 95% CI 0.76–0.94) [45]. The different toxicity profile of abiraterone and enzalutamide may assist during the treatment selection in some men with mCRPC, although they are both generally well tolerated and safe in the vast majority of patients.

#### 2.1.4. First-Line mCRPC in Patients Pretreated with ADT plus Docetaxel or ARSi

The choice of an optimal treatment for mCRPC patients who have received prior treatment with docetaxel for mHSPC remains unclear (Figure 1a). Data from the GETUG-AFU-15 trial showed that the benefit from docetaxel rechallange in mCRPC is limited in patients who have previously received docetaxel in mHSPC, as assessed by a PSA decline ≥50% obtained only in 14% of patients [46]. The significant OS benefit of abiraterone or enzalutamide in the phase III COU-301 and AFFIRM clinical trials in patients progressing after docetaxel in mCRPC seems to suggest that ARSi are a reasonable alternative in patients progressing after docetaxel for mHSPC. Cabazitaxel may, however, be an option in patients with adverse clinical features. In a recently published phase II study with patients with ARSi-naive mCRPC and poor prognosis features (presence of liver metastases, progression to mCRPC after <12 months of ADT, or ≥4 of 6 clinical criteria) who were allowed to receive docetaxel in mHSPC or mCRPC, cabazitaxel showed a greater clinical benefit compared to ARSi (80% versus 62%, *p* = 0.039) [47]. Patients who achieved stable disease for longer than 12 weeks were 75% for cabazitaxel and 56% for ARSi (*p* = 0.083), whereas there was no difference in terms of radiographic response rate or confirmed PSA decline ≥ 50%.

Chemotherapy appears to be a reasonable option for the first-line mCRPC treatment of eligible patients who have previously received ARSi in the mHSPC setting (Figure 1b) and in patients with nmCRPC who are progressing during treatment with ARSi. The extent of benefit is unknown due to the lack of prospective studies with this sequence, and clinical data of cross-resistance between ARSi and chemotherapy have been reported [48].

Cross-resistance between different ARSi is likely, and the sequence including two sequential ARSi is often discouraged. Although analyses from the SPARTAN trial in nmCRPC, where up to 80% of patients received abiraterone at progression, reported a benefit in PFS2 for patients in the apalutamide -> abiraterone over the placebo -> abiraterone sequence, most of the benefit was driven by the superior PFS of apalutamide over placebo in first-line nmCRPC, and the outcome comparisons in patients that received second-line therapy are lacking. Data from the control arm of the PLATO trial, in which patients received abiraterone acetate after first-line enzalutamide for mCRPC, are quite discouraging, with a median time to PSA progression of only 2.8 months and a PSA response ≥ 50% observed in 2% of patients [49]. A phase II crossover trial investigated the best sequence between abiraterone acetate -> enzalutamide (group A) vs. enzalutamide -> abiraterone acetate (group B) for the first-line treatment of 202 patients with newly-diagnosed mCRPC [50]. Longer time to PSA progression on second-line therapy (19.3 vs. 15.2 months, HR 0.66, 95% CI 0.45–0.97) and PSA response rates to second-line therapy (36% vs. 4%, *p* < 0.0001) were observed in patients treated with the abiraterone -> enzalutamide sequence, with no difference in OS (28.8 vs. 24.7 months, HR 0.79, 95% CI 0.54–1.16, *p* = 0.23).

Methodological issues arise when interpreting retrospective sequencing studies in mCRPC since, frequently, only results of patients treated in sequence are presented. The entire population of patients who start a first-line treatment should be analyzed to determine the best first-line approach in order to avoid a selection bias. The outcomes of patients with substantial benefit from first-line therapy and of those with aggressive disease that die while on first-line therapy, neither of which receive second-line treatment, can significantly affect the final results.

### 2.2. Selection of Subsequent Lines for mCRPC

Cabazitaxel (TROPIC trial), abiraterone acetate (COU-AA-301 trial), enzalutamide (AFFIRM), and radium-223 (ALSYMPCA trial) have demonstrated a significant improvement in OS after treatment with docetaxel in an mCRPC setting [21,23,24,51] (Table 1). No direct comparison among these agents is available. As previously mentioned, prospective data on the activity of either of these agents or docetaxel (a frequently used second-line agent after first-line hormonal agents) are limited.

Taken together, the data suggest the activity of agents in second-line is lower than in first line. The PSA response rates observed with enzalutamide in post-docetaxel mCRPC were lower than those observed in chemo-naïve mCRPC (78% vs. 54%) [23,52]. Similarly, the analysis of patients included in the COU-AA-302 trial who received docetaxel after abiraterone, consistently with different retrospective series, seems to suggest that the benefit of second-line docetaxel is lower than that observed in patients who received it in first-line [53,54]. Preclinical and clinical data suggest a variable degree of cross-resistance of abiraterone with enzalutamide but also of ARSi with docetaxel [48,55,56]; cabazitaxel, on the other hand, retains its clinical activity in patients pretreated with both chemotherapy and ARSi [57,58]. Retrospective data also support the notion that patients with early progression on first-line ARSi show increased response rates and time to PSA progression after treatment with second-line chemotherapy compared to the alternative ARSi [59].

The choice of therapy in patients that have received both an ARSi and docetaxel has been established in the phase III CARD trial, where cabazitaxel proved to be superior to a second ARSi [25]. In this study, 255 patients with mCRPC, who were previously treated with docetaxel and had progression within 12 months while receiving an ARSi (abiraterone or enzalutamide), received cabazitaxel or the alternative ARSi. Cabazitaxel showed significantly increased imaging-based PFS (HR 0.54, 95% CI 0.40–0.73) and OS (13.6 vs. 11.0 months HR 0.64, 95% CI 0.46–0.89), regardless of whether abiraterone or enzalutamide was received during the trial. Of note, the PSA response rates of a second ARSi after ARSi in the control arms of the CARD (13.5%) and PROFOUND (8%) trials are clearly inferior compared to those observed in post-docetaxel patients treated with abiraterone (38%) or enzalutamide (54%) in COU-AA-301 or AFFIRM [23,25,26,60].

### 2.3. Radiopharmaceutical Therapies

#### 2.3.1. The Role of Radium-223

Radium-223 is an intravenous alpha-emitting radiotherapeutic drug that mimics calcium and binds to bone mineral hydroxyapatite in areas of high bone turnover. In the phase III ALSYMPCA trial, six cycles of radium-223 at 50 kBq/kg prolonged OS (HR 0.70 95% CI 0.58–0.83) and delayed time to first symptomatic skeletal event (SSE) compared to placebo (HR 0.66 95% CI 0.54–0.77) in mCRPC patients with symptomatic bone metastases (no visceral disease, soft tissue disease > 2 cm or less than two bone metastases) (Table 1). Of note, only symptomatic pathologic bone fractures were included as SSE. Patients had either received docetaxel or were deemed ineligible or refused docetaxel; no patients had received abiraterone or enzalutamide [24]. Prior docetaxel was associated with higher rates of thrombocytopenia, but it did not appear to impair radium-223 efficacy [61]. A significant proportion of patients received docetaxel at progression, and chemotherapy after radium-223 was shown to be active with manageable side effects [62].

In the Expanded Access Program, the safety and activity of radium-223 was examined in a single-arm cohort of patients, including those with asymptomatic disease, and the combination of radium-223 with abiraterone or enzalutamide was allowed [63]. Patients receiving the combination of radium-223 with ARSi experienced a significantly longer OS compared to those receiving radium-223 alone. These results led to increased interest in the potential combinations of radium-223. However, the ERA-223 trial, a phase III randomized trial that compared abiraterone plus radium-223 with abiraterone alone in first-line mCRPC patients, was prematurely unblinded due to the high occurrence of bone fractures and deaths in the treatment arm of the trial. The combination of abiraterone and radium-223 was not shown to increase survival (HR 1.2, 95% CI 0.95–1.51). In addition, although the rate of SSE events was not different between arms, a higher rate of fractures (18% vs. 9%), mainly osteoporotic fractures (49% vs. 17%), was observed in the treatment arm. Of note, approximately 60% of patients included in the trial were not receiving bone protective agents [64]. These results led to the amendment of the other ongoing clinical trials such as the PEACE-3 phase III trial, comparing radium-223 plus enzalutamide with enzalutamide in first-line mCRPC, to mandate the use of bone protective agents in all patients. The use of bone protective agents significantly reduced the 12-month fracture incidence in patients treated with the combination (37.1% vs. 2.7%), and also in patients treated with enzalutamide alone (15.6% vs. 2.6%) [65]. According to the European Medicines Agency (EMA), the use of radium-223 is restricted for the treatment of men with mCRPC, symptomatic bone metastases, and no known visceral metastases, who are in progression after at least two prior lines of systemic therapy for mCRPC or ineligible for any available systemic mCRPC treatment [66]. Conversely, no restriction per line is included in the U.S. National Comprehensive Cancer Network Guidelines (NCCN). In view of the OS benefit with cabazitaxel as a third-line therapy in the CARD trial [25], radium-223 should be reserved as post-cabazitaxel therapy for patients with bone-predominant disease, unless deemed ineligible or refusing chemotherapy.

#### 2.3.2. The Advent of Lutetium-177-PSMA-617

Lutetium-177-prostate-specific membrane antigen (PSMA)-617 (LuPSMA) is an investigational radioligand therapy that has been investigated for patients with mCRPC [67]. LuPSMA binds with high affinity to PSMA, which is commonly expressed in prostate cancer including metastatic lesions, delivering β-particle radiation. The phase II TheraP trial enrolled 200 patients with mCRPC for whom cabazitaxel was considered the next appropriate standard treatment [68]. The PET eligibility criteria for the trial were PSMA-positive disease and no sites of metastatic disease with discordant FDG-positive and PSMA-negative findings. Of note, about 1/3 of patients who had registered for the study (91/291) were ineligible prior to randomization either because of low PSMA expression or FDG discordant disease. Compared with cabazitaxel, Lu-PSMA led to a higher PSA response (66% vs. 37%, *p* < 0.0001) and fewer grade 3 or 4 adverse events (33% vs. 53%).

The results of the phase 3 VISION study involving patients with mCRPC treated with LuPSMA were recently presented at the ASCO Congress 2021 [27] (Table 2). In this study, men previously treated with at least one ARSi and one taxane were randomized to receive LuPSMA plus standard of care vs. standard of care alone. Eligible patients had at least one PSMA-positive metastatic lesion and no PSMA-negative metastatic lesions. PSMA criteria were met in 86.6% of patients. Compared to standard-of care alone, LuPSMA significantly prolonged OS (median 15.3 vs. 11.3 months, HR 0.62 95%CI 0.52–0.74) and radiographic PFS (median 8.7 vs. 3.4 months, HR 0.40 99.2% CI 0.29–0.57). Overall, this treatment was safe and tolerable. Of note, standard of care in the control arm excluded chemotherapy, immunotherapy, radium-223, and investigational drugs, which led to a very high (56%) initial drop-out rate in the control arm before receiving treatment. Based on these data, Lu-PSMA can be considered an option for patients that have exhausted all active lines of therapy and present PSMA uptake in PET-scans. The fact that more than 85% of patients in the VISION trial met the PSMA criteria has raised the question whether it would be reasonable to use Lu-PSMA therapy on the sole basis of standard imaging [69]. However, the TheraP trial required an FDG-PET to exclude patients with metabolically active, probably low-differentiated disease sites lacking PSMA expression; patients’ outcomes, not surprisingly, appeared superior to those reported in the VISION study and might serve as a further argument for further optimizing the eligibility screening. Of note, Lu-PSMA is also being prospectively evaluated as metastasis-directed therapy after surgery and external beam radiotherapy in patients with low-volume mHSPC [70].

### 2.4. Bone-Targeted Therapies

Given the high prevalence of bone metastases, bone resorption inhibitors (BRI) have emerged as potential options for the prevention of SRE among men with prostate cancer. Zoledronic acid and denosumab have demonstrated the ability to reduce the risk of skeletal-related events (SRE)—including asymptomatic fractures—and time to first SRE in men with mCRPC [71,72]. Of note, these trials have been conducted before the advent of ARSi and radium-223 that have been also shown to prevent SRE. In addition, none of these agents has ever demonstrated an OS benefit in a randomized trial. However, several retrospective data support the notion that the addition of BRI to contemporary therapies might prolong survival [73,74]. International guidelines recommend in favor of their use in patients with mCRPC, although their potential toxicity (e.g., osteonecrosis of the jaw, hypocalcaemia) must always be kept in mind. Importantly, in men with mHSPC, treatment with zoledronic acid was not associated with a lower risk for SRE, and the use of BRI in this early setting is not sustained by clinical evidence [75].

### 2.5. Treatment Combinations

In an attempt to maximize benefits, a number of combinations of agents with seemingly non-overlapping mechanisms of action have been studied in advanced prostate cancer [76]. Combinations, for instance, of different ARSi with chemotherapy in mHSPC have been pursued, with conflicting results.

In the ENZAMET trial, the use of enzalutamide in combination with docetaxel was associated with significant improvement in clinical PFS (HR 0.48 95% CI 0.37–0.62), but the hazard ratio was suggestive for no OS benefit (HR 0.90, 95% CI 0.62–1.31). Of note, no evidence of heterogeneity of effect according to docetaxel use was found (adjusted *p* = 0.14), and this result should be interpreted with caution. Similar data were observed in the post-hoc analysis of the TITAN trial of apalutamide in mHSPC [7]. Only 11% of patients had received prior treatment with docetaxel, and such subgroup analyses are purely exploratory. In these patients treated with chemotherapy, the benefit of adding apalutamide was consistent with the overall population in terms of radiographic PFS (HR 0.47 95% CI 0.22–1.01), but it was unclear in terms of OS (HR 1.27 95% CI 0.52–3.09). The ARASENS trial, a randomized, double-blind, placebo-controlled, phase III trial, is currently evaluating the AR antagonist darolutamide plus standard ADT plus docetaxel [77].

The recently presented results of the PEACE-1 trial also confirmed the potential benefit of adding abiraterone acetate to docetaxel in men with mHSPC in terms of radiographic PFS (HR 0.50 95% CI 0.40–0.62) [78]; data on OS are awaited before the clinical relevance of this combination can be established. This trial will also provide information about the addition of local radiotherapy to abiraterone acetate in mHSPC. Currently, it remains uncertain whether patients with low-volume mHSPC who start an ARSi should also receive radiotherapy to the primary tumor. In a recent Twitter survey from the Advanced Prostate Cancer Consensus Conference 2021, 76% of 144 respondents would recommend adding local RT to apalutamide or enzalutamide in low volume mHSPC, even if there is no scientific evidence to date regarding this combination approach.

In the mCRPC setting, two phase III trials evaluated the combination of abiraterone with the antiandrogens enzalutamide (ALLIANCE A031201) and apalutamide (ACIS trial) compared with ARSi alone as first-line mCRPC treatment. Both abiraterone plus enzalutamide (HR: 0.70 95% CI 0.67–0.72) and abiraterone plus apalutamide (HR: 0.69, 95% CI 0.58–0.83) showed a significant benefit in terms of radiographic PFS over ARSi monotherapy but no OS benefit [79,80]. The combination of enzalutamide and docetaxel was shown to increase PFS over docetaxel alone as first line-therapy for mCRPC in the phase II CHEIRON trial [81]. Currently, the randomized phase II CHAARTED2 trial is actively recruiting mCRPC patients who received prior docetaxel chemotherapy for high volume mHSPC to receive abiraterone acetate with or without cabazitaxel [82]. In the recently presented IPATENTIAL 150 phase III study, the combination of abiraterone and the PI3K inhibitor ipatasertib was shown to increase radiographic PFS compared to abiraterone alone as first-line mCRPC therapy in patients with loss of PTEN; OS data are awaited to define the role of this combination in the treatment of mCRPC [20]. A number of different combinations of hormonal and chemotherapeutic agents with other agents such as radiopharmaceuticals (radium-223), PARP inhibitors (olaparib), or immunotherapeutic agents (nivolumab, pembrolizumab) have reported clinical activity in mCRPC [83,84,85,86]. However, to date, none of these combinations have provided evidence of an OS benefit in randomized trials, and their use cannot be recommended as standard of care outside clinical trials.

## 3. Predictive Biomarkers and Potential Impact on Treatment Sequence

Several biomolecular alterations, including alterations in tumor driving genes, have been observed in patients with prostate cancer. Some of these molecular alterations could be explored as predictive biomarkers for planning treatment to early identify primary resistance, avoiding useless toxicity to patients. In some cases, these alterations involve inherited or spontaneously acquired gene mutations in the germline. More frequently, alterations are acquired at the somatic level during the oncogenesis and/or cancer progression, or they could arise or be enriched as a result of the selective pressure induced by treatments. Examples of molecular alterations associated with the mechanisms of treatment resistance that could be helpful in castration-resistant disease to select the appropriate therapy include androgen receptor (AR) amplification, mutation, or splice variants. Other resistance mechanisms bypass AR by exploiting alternative signaling and metabolic pathways [87]. Table 2 summarizes the evidence for proposed molecular biomarkers in advanced prostate cancer. Some DNA damage and response genes (DDR), especially *BRCA1/2,* have been clinically validated as biomarkers for selecting patients who are sensitive to poly ADP-ribose polymerase (PARP) inhibition. Pembrolizumab has received tissue-agnostic approval by the U.S. Food and Drug Administration (FDA) for patients with microsatellite instability or mismatch repair-deficient prostate cancer. In addition, the *AKT* inhibitor ipatasertib has demonstrated significant activity in patients with *PTEN* loss. Many of these biomarker-driven treatments are going to be implemented in routine clinical practice. However, to what extent these treatments will affect the sequencing and response of other therapies is largely unknown and will be the object of investigation in the future. 

### 3.1. DDR Genes

Alterations in DDR genes have recently become a field of major interest in prostate cancer research, given their potential prognostic and predictive implications [88,89]. DDR defects have been encountered in the germline of 8–17% of patients with metastatic disease [90,91,92]. *BRCA2* gene alterations are the most common DDR event both in the somatic- and germline [90,93].

Germline *BRCA2* mutations have been associated with aggressive disease and poor clinical outcomes [94,95]. The PROREPAIR-B study has shown that the detection of germline *BRCA2* alterations has negative prognostic significance. Additionally, a significant interaction between germinal *BRCA2* status and treatment type (ARSi versus taxane therapy) has been observed, suggesting that *BRCA2* might be a valid biomarker during the selection of the first-line treatment choice in patients with mCRPC [90]. The BRCA2men study aims to validate germline *BRCA2* alterations as a predictive biomarker for the selection of ARSi or taxanes as first-line of therapy [96].

**Table 2 cancers-13-04522-t002:** Promising predictive biomarkers in mCRPC.

Biomarker	Source	Drugs	Studies	Phase III Trials
**DDR** (*BRCA1/2*, *ATM*, *PALB2* and other genes)	PMBC, tumor tissue or ctDNA	Olaparib Rucaparib Talazoparib Niraparib	Phase 2 TOPARP [97] Phase 2 TRITON-2 [98] Phase 2 TALAPRO-1 [99] Phase 2 GALAHAD [100]	PROFOUND [26,83] PROpel [101] * KEYLINK-010 [102] * TRITON-3 [103] * CASPAR [104] * TALAPRO-2 [105] * MAGNITUDE [106] *
* **PTEN** * **loss**	Tumor tissue	Ipatasertib Capivasertib	Phase 2 A. Martin study [107] Phase 2 ProCAID [108]	IPATential150 [109]
**AR-V7**	CTCs	ARSi	PROPHECY biomarker study [110]	
**Molecular subtype** Luminal A Luminal B Basal	Tumor tissue	Apalutamide Docetaxel	SPARTAN [111] and TITAN [112] (biomarker analyses) CHAARTED [113] (biomarker analysis)	
**Others** MSI-h/MMRd *CDK12* deficiency *SPOP* mutations *RB1* loss *TP53* alterations *TMPRSS2*	Tumor tissue	ARSi ICI	*Explorative analyses*	

ARSi: androgen receptor signaling inhibitors; AR-V7: androgen-receptor variant 7; CTC: circulating tumor cells; ctDNA: circulating tumor DNA; DDR: DNA damage response (genes); ICI: immune checkpoint inhibitors; mCRPC: metastatic castration-resistant prostate cancer; MSI-h/MMRd: microsatellite instability-high/mismatch repair deficient; PBMC: peripheral blood mononuclear cells. * Ongoing trials.

Platinum-based chemotherapy represents one of the first fields of investigation in patients with prostate cancer harboring DDR defects. Platinum generates DNA crosslinks that cannot be easily repaired when the homologous recombination repair (HRR) pathway is impaired, leading to cell death. This strategy has proven successful in treating breast and ovarian cancers with alterations in *BRCA1* or *BRCA2*. Several case series and retrospective studies suggest that DDR-deficient prostate cancer patients might benefit from this therapeutic approach, and many clinical trials are ongoing to assess the role of platinum-based chemotherapy in patients with DDR defects [88].

Practice-changing data came from trials including patients with DDR defects treated with PARP inhibitors. The phase III PROFOUND study has recently established the predictive value of certain DDR genes defects in patients with mCRPC whose disease had progressed during previous treatment with enzalutamide, abiraterone, or both [26,83]. Patients that had progressed on one prior ARSI were randomized to receive olaparib or the physician’s choice of enzalutamide or abiraterone (control). 65% of patients had also received prior taxane therapy. Treatment with olaparib significantly prolonged the PFS and OS of patients with at least one alteration in *BRCA1*, *BRCA2*, or *ATM*, establishing the first validated biomarker in patients with prostate cancer.

The subgroup analysis of PFS and OS favored olaparib irrespective of prior taxane use [114]. The use of a second ARSI as a suboptimal control arm is a potential limitation for the interpretation of the PROFOUND study results; PSA and objective response rates in the control group were of only 10% and 4%, respectively. It must be noted, however, how over 66% of patients progressing on the control arm crossed over to receive olaparib upon disease progression. Results from the PROFOUND trial established olaparib as the standard of care in patients with DNA repair alterations progressing on prior ARSI with and without prior chemotherapy. It is unknown, however, whether olaparib provides greater activity than cabazitaxel in this setting, based on the similar population treated in the CARD trial [25]. Indirect comparisons between PROFOUND and CARD trials suggest that patients harboring BRCA1/2 or ATM alterations treated with olaparib show improved radiographic PFS compared to those treated with cabazitaxel [115,116]. Conversely, cabazitaxel seems to outperform in patients with other HRR variants [115]. The phase II TRITON 2 trial established the activity of the PARP inhibitor rucaparib in patients with mCRPC and *BRCA1/2* alterations who had progressed after one to two lines of ARSi and one taxane-based chemotherapy for mCRPC, with complete response rates and a confirmed PSA response rate of 43.5% and 54.8%, respectively [98].

Current evidence suggests that different DDR alterations could provide different sensitivity to PARP inhibitors. In the PROFOUND trial, the gene subgroup analysis suggested that patients with *BRCA* alterations are those who derive the greatest benefit from olaparib, whereas those with *ATM* alterations showed unclear PFS (HR: 1.04, 95% CI 0.61–1.87) and OS benefit (HR: 0.93, 95%CI 0.53–1.75) [117]. In the phase II, single arm TOPARP-B trial [118], *BRCA1/2* germline and somatic pathogenic mutations were associated with similar benefit from olaparib; greater benefit was observed in patients with homozygous *BRCA* deletion. Biallelic, but not mono-allelic, *PALB2* deleterious alterations were associated with clinical benefit. In addition, the loss of ATM protein by immunohistochemistry was associated with a better outcome. Of note, the loss of RAD51 foci, a functional biomarker of HRR function, was primarily found in tumors with biallelic BRCA1/2 and PALB2 alterations, and the authors have suggested that the RAD51 assay could help identify less-common genomic variants impacting HRR function that sensitize to PARP inhibition.

In the TRITON2 trial, PSA response rates were greater in patients with germline versus somatic *BRCA1/2* mutations, in biallelic versus monoallelic mutations, and in homozygous deletions versus other deleterious mutations. In addition, the efficacy of rucaparib was greater in patients with *BRCA2*- versus *BRCA1*-altered mCRPC, as assessed by PSA50 response rates, overall response rates, and median radiographic PFS estimates. This apparent discrepancy in PARP inhibitor sensitivity between patients with *BRCA1*- and *BRCA2*-mutated mCRPC seems to be a class effect of PARP inhibitors in prostate cancer [119]. Taza and colleagues found that PARP inhibitor activity was diminished in *BRCA1*- versus *BRCA2*-altered mCRPC in a cohort of 123 *BRCA1/2*-altered mCRPC patients receiving the PARP inhibitor, and this differential activity was not explained by mutation origin (germline vs. somatic) or allelic status (mono- vs. biallelic) [120]. The phase II TALAPRO-1 trial reported results from the treatment with talazoparib in patients with mCRPC and associated DDR defects who had progressed after ARSi and taxane [99]. The overall response rates were 44% in patients harboring *BRCA1/2* alterations, 33% in *PALB2* and 12% in *ATM,* whereas the complete response rates were 76% in *BRCA1/2*, 50% in *PALB2*, and 28% in *ATM.* The phase II GALAHAD trial is assessing niraparib in patients with mCRPC and biallelic DDR defects with disease progression on taxane and ARSi [100]. At the interim analysis, niraparib showed an overall response rate of 41% and a complete response rate of 63% in *BRCA* carriers, with durable responses, particularly in biallelic BRCA mutation carriers.

We could conclude that olaparib and other PARP-inhibitors as monotherapy showed significant benefit in patients with pretreated mCRPC and alterations in DDR, especially in those with *BRCA1/2* alterations. The clinical use of these agents is dependent on local regulatory approval; for example, the U.S. FDA approved olaparib for men with deleterious or suspected deleterious germline or somatic HRR gene-mutated mCRPC who have progressed on ARSi. In contrast, the EMA have restricted its use to patients with germline or somatic BRCA1/2 mutations.

Ongoing studies are assessing the role of these agents in combination with ARSi at earlier stages of mCRPC, given the strict relationship between PARP1 activity and AR function. It is also hypothesized that the co-blockade of PARP1 and AR using could be active regardless of DDR deficiency status. A phase II trial of olaparib in combination with abiraterone in post-docetaxel mCRPC showed a significant improvement in terms of radiographic PFS with the combination compared to abiraterone alone [121]. The ongoing PROpel Phase III trial is testing olaparib as a first-line treatment for patients with mCRPC in combination with abiraterone versus abiraterone alone irrespective of DDR status, and this could extend the use of these agents in unselected populations of patients with mCRPC [101]. Other clinical trials are testing other PARP-inhibitors in combination with ARSi for the first-line treatment of mCRPC (Table 2).

### 3.2. AR Pathway

Several studies support the notion that alterations in the AR pathway represent an important driver of resistance in the context of mCRPC. AR aberrations including point mutations, copy number variations (CNV), structural variations, and alternatively spliced forms of AR are frequent among mCRPC patients, particularly after the use of ARSi [122]. An analysis of plasma cell-free DNA (cfDNA) showed that the detection of AR amplification and heavily mutated AR are associated with worse PFS in patients with mCRPC treated with enzalutamide [123]. Circulating *AR* CNV in plasma DNA are associated with a worse outcome in patients with mCRPC treated with ARSi [124]. *AR* gain in plasma DNA is also associated with a worse outcome in docetaxel-treated mCRPC patients, but *AR*-gained patients seem to derive greater benefit from treatment with taxanes than with ARSi [125,126].

The androgen-receptor variant 7 (AR-V7) has been proposed to predict for poor response to treatment with ARSi, such as abiraterone acetate or enzalutamide. Antonarakis and colleagues firstly showed that the detection of this AR variant in circulating tumor cells (CTCs) was associated with treatment resistance to ARSi [127]. Interestingly, AR-V7 did not seem to be associated with resistance to taxane-based chemotherapy, and the potential reversion of AR-V7 detection was observed after taxane treatment [128,129,130]. In the PROPHECY trial, 118 men with mCRPC who were starting abiraterone or enzalutamide were enrolled to assess the role of AR-V7 [110]. AR-V7 detection by both the Johns Hopkins and Epic AR-V7 assays was independently associated with shorter PFS and OS, and patients with AR-V7–positive mCRPC had fewer confirmed prostate-specific antigen responses or soft tissue responses. However, no randomized trial has ever demonstrated that alternative treatment with chemotherapy in AR-V7–positive patients could clearly translate into a survival benefit, and the potential confounding prognostic effects of AR-V7 have called into question its predictive value and its clinical utility. AR-V7 is rarely detected in patients who are starting a first-line treatment for mCRPC after androgen-deprivation therapy (3–8%). In the ARMOR3-SV trial, AR-V7 was detected in only 8% of 953 men with treatment-naïve mCRPC [131]. However, the prevalence of AR-V7 progressively increases with the number of treatment lines received for mCRPC [132,133]. The NCCN guidelines state that AR-V7 testing can be considered to help guide the selection of therapy in the post-ARSi mCRPC setting [134]. However, its clinical use outside of a clinical trial should be discouraged unless a randomized study confirms its predictive role. Based on the results of the CARD trial, cabazitaxel should be the standard of care in patients who had received prior docetaxel and are progressing during ARSi, irrespective of AR-V7 status. AR-V7 assessment may become useful in those patients who are not eligible or are not prone to chemotherapy to inform them that a second treatment with ARSi may be ineffective.

### 3.3. PTEN Loss and PI3K Alterations

About a half of patients with mCRPC show a loss of the *AKT* phosphatase *PTEN*, with hyper-activation of the oncogenic *PI3K/AKT* signaling [135]. These patients show worse prognosis and reduced benefit from treatment with ARSi [136]. The phase II A. Martin study assessed the activity of the *AKT* inhibitor ipatasertib plus abiraterone vs. abiraterone alone in patients with mCRPC after docetaxel chemotherapy [107]. The radiographic PFS was prolonged in the ipatasertib cohort, with similar trends in OS and time-to-PSA progression; in addition, a larger radiographic PFS prolongation for the combination was demonstrated in *PTEN*-loss tumors. Based on these data, the phase III IPATential150 trial assessed the efficacy ipatasertib in combination with abiraterone compared to abiraterone alone for the first-line treatment of patients with mCRPC [109,137]. The co-primary endpoints were radiographic PFS in the *PTEN*-loss-by-immunohistochemistry population and in the intention-to-treat population. Of 1101 patients enrolled in this study, 521 (47%) harbored *PTEN* loss. In patients with *PTEN* loss, the combination arm with ipatasertib achieved significantly superior radiographic PFS (18.5 vs. 16.5 months, HR 0.77, 95% CI 0.61–0.98, *p* = 0.034) and antitumor activity compared to the placebo arm. However, the improvement of radiographic PFS in the intention-to-treat (ITT) population was not statistically significant. The subgroup analysis of the IPATential150 trial suggests that patients with *PTEN* loss previously treated with taxanes may not benefit from the addition of ipatasertib to abiraterone (HR 1.0 95% CI 0.58–1.74). However, given the limited number of patients, this observation should be interpreted with caution. A biomarkers analysis of the IPATential150 trial also showed that patients with *PTEN* loss and with genomic alterations in *PIK3CA/AKT1/PTEN* by next generation sequencing had a larger magnitude of radiographic PFS benefit with ipatasertib than patients with no detectable alterations [138]. These results support the notion that ipatasertib plus abiraterone is a valid treatment option for first-line mCRPC with *PI3K/AKT* pathway alterations. The ProCAID phase 2 trial assessed the efficacy of docetaxel combined with pan-*AKT* inhibitor capivasertib compared to docetaxel alone in patients with mCRPC. The primary endpoint of PFS was not met, irrespective of the biomarker status for the PI3K/AKT/PTEN signaling pathway. However, OS (secondary endpoint) was longer in patients who received the combination compared with chemotherapy alone, and prospective validation studies are required to identify patients most likely to benefit from capivasertib [108].

### 3.4. Basal Versus Luminal Prostate Cancer

The PAM50 is a well-known gene expression classifier that categorizes breast cancer into luminal A, luminal B, HER2, and basal subtypes. Zhao and colleagues applied this classifier to subtype prostate cancer samples into luminal A, luminal B, and basal subtypes [139]. The authors found that luminal B prostate cancers had the poorest clinical outcomes, followed by basal, and luminal A. Although both luminal-like subtypes were associated with increased AR expression and signaling, only luminal B prostate cancers were significantly associated with postoperative response to ADT. Similar results were observed with chemotherapy in patients included in the CHAARTED trial [113]. In the control arm with ADT alone, the luminal B subtype was associated with shorter OS compared to the basal subtype, confirming the negative prognostic significance of the luminal B subtype. However, patients with the luminal B subtype treated with ADT plus docetaxel showed significant improvement in time to castration-resistance and OS, whereas the basal subtype showed no OS benefit from ADT plus docetaxel, including in patients with high-volume disease. The luminal subtype also seems to respond better to ARSi compared to the basal subtype.

Regardless of basal/luminal subtype, >50% of patients enrolled in the phase III SPARTAN trial (apalutamide in nmCRPC) achieved ≥90% reduction in PSA with apalutamide. However, PSA decline was deepest and most rapid in patients with the luminal subtype. Similarly, the OS improvement with apalutamide seemed to be more marked in patients with the luminal subtype (HR 0.43, 95% CI 0.19–1, *p* = 0.051) compared to the basal subtype (HR 0.67, 95% CI 0.40–1.14, *p* = 0.14) [111]. Conversely, in the sub-analysis of the TITAN trial (apalutamide in mHSPC), the prolongation of radiographic PFS induced by apalutamide seemed to be more sustained in the basal molecular subtype (HR 0.31 95% CI 0.16–0.62, *p* = 0.0008) compared to the luminal subtype (HR 0.74, 95% CI 0.40–1.36, *p* = 0.33) [112].

It is unclear whether statistical inadequacy or the distinct setting (mHSPC vs. mCRPC) might explain these discordant results. Of note, an increased proportion of patients with the basal subtype was found in SPARTAN compared to TITAN (66% vs 50%). However, biomarkers analyses were performed in archival primary tumors, including patients that received these treatments in later stages during castration-resistance. The molecular characteristics of metastatic sites might differ from those of primary tumors; therefore, caution should be used when interpreting these analyses. Overall, these data suggest that luminal versus basal classification might be useful to selecting patients who are expected to derive the greatest benefit from ARSi and docetaxel. However, prospective biomarker-driven studies are needed to determine the real potential predictive impact of this classification.

### 3.5. Aggressive-Variant Prostate Cancer

Aggressive-variant prostate cancer (AVPC) refers to AR-independent anaplastic forms of prostate cancer that are characterized by a rapidly progressive disease, weak response to therapies, and poor prognosis [140]. Many of these tumors are prostate cancers with neuroendocrine features (NEPC), but some of these cases do not show the typical morphology or immunohistochemical profiles of neuroendocrine differentiation. AVPC cells can arise de novo or, more commonly, be the result of divergent clonal evolution from one or more castration-resistant adenocarcinoma cell [141]. The selective pressure induced by chemotherapy and ARSi favors the emergence of such resistant clones, which are commonly found in the advanced stages of castration-resistance. The loss of *RB1* and *PTEN*, *TP53* mutations, and the amplification of *MYCN* and *AURKA* are common events in NEPC and AVPC [142,143]. AVPC is characterized by clinical characteristics of aggressiveness, such as histologic evidence of NEPC, the presence of exclusively visceral metastases, predominant lytic bone metastases, bulky disease, or low PSA at initial presentation with high volume bone metastases [140,144,145].

NEPC generally shows a high response rate, generally of short duration, to platinum-based chemotherapy [144]. These patients are unlikely to respond to ARSi [146], and the NCCN guidelines currently recommend using chemotherapy with cisplatin/etoposide, carboplatin/etoposide, and docetaxel/carboplatin as first or subsequent treatments for patients with small-cell or NEPC [134]. A phase II study investigated the use of the *AURKA* inhibitor alisertib in patients with metastatic NEPC [147]. Although the trial did not meet its primary endpoint of improved PFS, the tumors suggestive of *N-myc* and *Aurora-A* overactivity showed exceptional responses, including the complete resolution of liver metastases and prolonged stable disease. Many trials are currently ongoing in patients with AVPC and NEPC to test the activity of immunotherapy, PARP inhibitors, and EZH2 inhibitors in these patients [148].

For patients with AVPC (excluding those with small-cell or NEPC histology) there is no consensus for the optimal first-line treatment. At the Advanced Prostate Cancer Consensus Conference (APCCC) 2019, 75% of panelists voted to add docetaxel to ADT, 16% voted to add platinum-based combination therapy, and 9% voted to add an ARSi. Finally, the potential effect of a first-line platinum-based chemotherapy on the efficacy of subsequent treatments such as PARP inhibitors, docetaxel, or ARSi is largely unknown and requires further studies.

### 3.6. Other Molecular Biomarkers

Given its tissue-agnostic approval by the FDA, patients with microsatellite instability or mismatch repair-deficient prostate cancer tumors might benefit from treatment with pembrolizumab [149]. In the study by Abida and colleagues, among 1033 patients who had adequate tumor quality for microsatellite instability (MSI) analysis, 32 (3.1%) had MSI-high/mismatch-deficient prostate cancer and seven of them had a pathogenic germline mutation in a Lynch syndrome-associated gene [149]. Six of eleven patients (54.5%) who received anti-programmed cell death protein 1 (PD1)/ligand 1 (PD-L1) therapy had a >50% decline in PSA levels, and four of them had radiographic responses. However, none of the six patients with tumor response included in the Phase II KEYNOTE-199 study of pembrolizumab in mCRPC were found to have microsatellite instability, suggesting that other mechanisms could be also involved in favoring response to immunotherapy [84]. Of interest, 2/19 patients (11%) with *BRCA* or *ATM* aberrations included in this trial showed response to pembrolizumab, compared to 4/124 (3%) of those without alterations in DDR. The data also suggest that a proportion of patients with *CDK12* deficiency may respond favorably to anti-PD-1 checkpoint inhibitors [150,151]. *SPOP* mutations have been suggested to predict for response to abiraterone acetate [152]. *RB1* aberrations increase in prevalence after treatment-selective pressure [153]; patients with mCRPC treated with enzalutamide and concurrent *RB1* alterations showed worse clinical outcomes and worse progression-free survival [123]. A study also found that alterations in *RB1* and *TP53* are associated with shorter time on treatment with abiraterone or enzalutamide [154]. Another study also suggested that the cooperative loss of two or more tumor suppressor genes, including *TP53, PTEN,* and *RB1*, may drive more aggressive disease and an increased risk of relapse [155].

### 3.7. Molecular Biomarkers and Diagnostic Challenges

Of 4425 patients initially enrolled in the PROFOUND trial, 4047 patients had tumor tissue available for testing. Among these, 2792 (69%) were successfully sequenced, and only 162 patients (3.7% from initial enrollment) were found to harbor germline or somatic alterations in these *BRCA1*, *BRCA2*, or *ATM*. These data show the important limits of tumor tissue analysis. An increase in the sequencing success rate or the implementation of liquid biopsy approaches are necessary to enlarge the number of patients who could benefit from biomarker-driven treatments. It has been shown that ctDNA can sufficiently identify all driver DNA alterations found in matched metastatic tissue in the majority of patients with mCRPC [156]. Data from the PROFOUND trial found a high concordance between tumor tissue and circulating tumor DNA (ctDNA), supporting the development of ctDNA testing as a minimally invasive method to identify patients with DDR-altered mCRPC [157]. In metastatic disease, ctDNA can identify somatic mutations, copy-number variations, and structural rearrangements that are predictive of response to therapies. However, multiple technical and biological variables can confound the ctDNA-based genotyping, complicating the implementation of ctDNA into clinical practice [158]. The ctDNA fraction (ctDNA%) strongly influences assay detection sensitivity and specificity for different genomic events, and it is a critical variable during the interpretation of patient results. For example, the copy number variations in *TP53*, *BRCA2*, *PTEN*, *RB1*, and *AR* all have clinical relevance in mCRPC, but these alterations are not always possible to identify in samples with low ctDNA% [158]. Importantly, dynamic changes in gene mutational status have been observed in same-patient samples between hormone-naive and mCRPC biopsies [159]. This observation highlights that biopsies performed at initial diagnosis do not necessarily reflect the tumor mutational status of the advances stages of castration-resistance. Therefore, both ctDNA and tumor tissue analysis show advantages and constraints and are likely to become more complementary than competing in the era of precision oncology. The development of more accurate and feasible assays to easily detect the presence of specific biomolecular alteration in patients with cancer will be the challenge of the next decades.

### 3.8. PET Tracers as Predictive Biomarkers

Given that the majority of pivotal trials in mHSPC and mCRPC have been conducted using standard imaging for staging, limited evidence is available regarding the potential predictive role of PET tracers during the staging and re-staging of patients with prostate cancer. Several PET-derived parameters might be of value for the prognostic stratification of patients with mCRPC before systemic therapy [160]. In addition, PET imaging might better reflect treatment response and may allow one to avoid useless toxicity in resistant patients and switch them earlier to more effective therapeutic options. As previously mentioned, data from the TheraP and VISION trials support the notion that patients with significant PSMA-PET uptake are those expected to derive the greatest benefit from Lu-PSMA [27,68]. Some studies suggest that FDG- and choline-PET can adequately identify patients who better respond to treatment with ARSi and radium-223 in the mCRPC setting [161,162,163]. However, prospective studies are warranted to determine that the early identification of response/progression—not detected by standard imaging—is clinically useful and could affect the prognosis of patients. For example, the sole PSA rise, which might be an early signal of cancer progression, is not a reason to discontinue therapy until radiographic or clinical progression is manifest [164,165]. Therefore, caution should be used when interpreting progression according to these novel techniques, especially in the mCRPC setting.

## 4. Conclusions

In recent years, new standards of care have been established for the treatment of advanced prostate cancer. However, few randomized trials have investigated which might be the best sequencing approach among these options in order to maximize the benefit and prolong the survival of patients. The choice of first-line treatment for mCRPC is complicated by the introduction of treatment options at the earlier stages of mHSPC and nmCRPC. The CARD trial has defined that chemotherapy with cabazitaxel is the best choice for patients with mCRPC who have already received docetaxel and are progressing during an ARSi. The ARSi -> ARSi approach is commonly discouraged given the potential development of cross-resistance. The advent of precision treatments, such as PARP- and AKT-inhibitors, and the pending approval of LuPSMA are going to further complicate this complex scenario. In the future, randomized trials are warranted to identify the optimal sequencing strategy and to improve the outcomes of patients with both hormone-sensitive and castration-resistant disease.

## Figures and Tables

**Figure 1 cancers-13-04522-f001:**
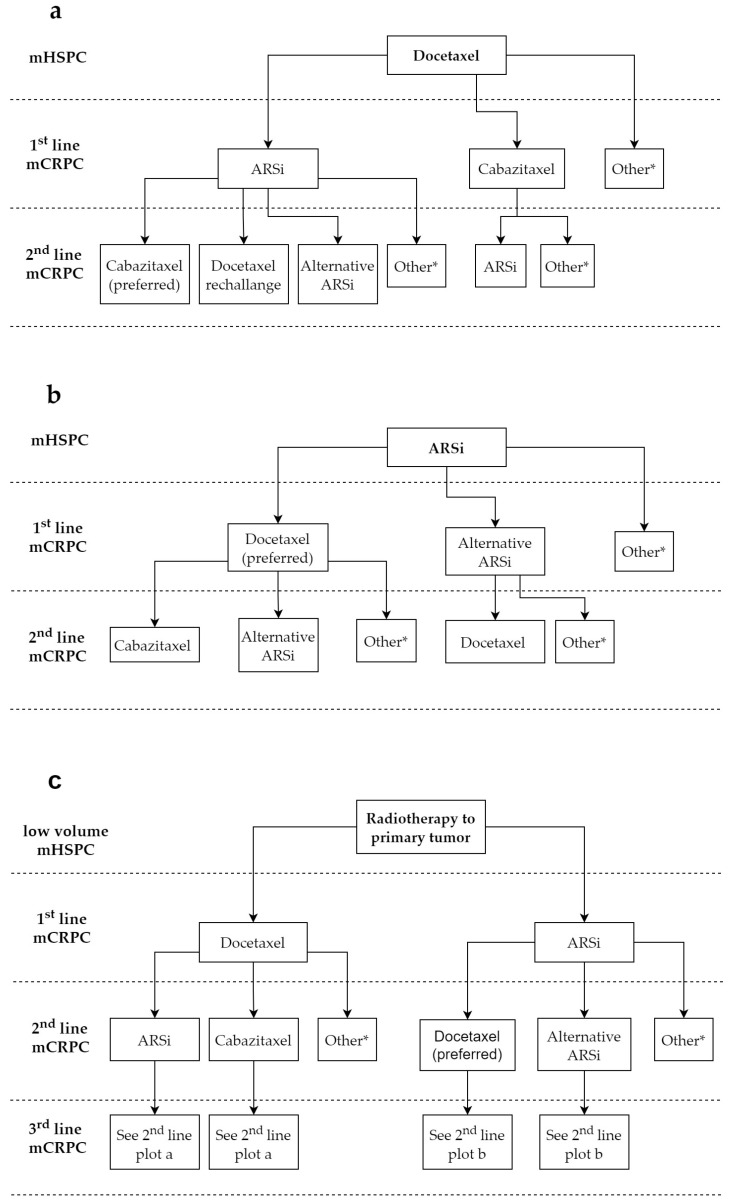
Possible sequencing scenarios in patients with mHSPC initially treated with: (**a**) Docetaxel; (**b**) ARSi; (**c**) Radiotherapy to the primary tumor. Patients with small-cell or neuroendocrine prostate cancer should start a first-line treatment with platinum-based chemotherapy. ARSi: androgen-receptor signaling inhibitors. * Other treatments can include: radium-223 for patients with symptomatic bone metastases; lutetium-177-prostate-specific membrane antigen (PSMA)-617 (LuPSMA) after regulatory approval; olaparib or other PARP-inhibitors for patients with DNA damage and response (DDR) genes defects; pembrolizumab or other immunotherapy for patients with microsatellite instability (MSI)-high/mismatch-deficient prostate cancer; sipuleucel-T (only USA); and mitoxantrone for palliation in symptomatic patients who cannot tolerate other therapies.

**Table 1 cancers-13-04522-t001:** Prospective randomized clinical trials in mHSPC, nmCRPC, and mCRPC.

Setting	Name of the Trial	Population	Exp Arm	Control Arm	N Exp/ Cont	Primary Endpoint	FU (months)	mOS (months) Exp/Contr	HR (95% CI)	Ref.
**mHSPC**	**GETUG-AFU 15**	L/H volume	Doce + ADT	ADT	192/193	OS	83.9	62.1/48.6	0.88 (0.68–1.14)	[9]
	**CHAARTED**	L/H volume	Doce + ADT	ADT	397/393	OS	53.7	57.6/47.2	0.72 (0.59–0.89)	[1]
	**STAMPEDE (arm C)**	L/H volume	Doce + ADT	ADT	362/724	OS	78.2	59.1/43.1	0.81 (0.69–0.95)	[2]
	**ENZAMET**	L/H volume	Enza + ADT ± doce	NSAA + ADT ± doce	563/562	OS	34	NE/NE	0.67 (0.52–0.86)	[6]
	**ARCHES**	L/H volume; prior doce allowed	Enza + ADT	Placebo + ADT	574/576	rPFS	14.4	NE/NE	0.81 (0.53–1.25)	[10]
	**TITAN**	L/H volume; prior doce allowed	Apa + ADT	Placebo + ADT	525/527	OS and rPFS	44	NE/52.2	0.65 (0.53–0.79)	[11]
	**LATITUDE**	High risk	AA + P + ADT	Placebo + P + ADT	597/602	OS	51.8	53.3/36.5	0.66 (0.56–0.78)	[4]
	**STAMPEDE (arm G)**	L/H risk L/H volume	AA + P + ADT	Placebo + P + ADT	501/502	OS	73.2	79.2/45.6	0.60 (0.50–0.71)	[5]
	**STAMPEDE** **(arm H)**	L/H volume	RT to prostate + ADT	ADT	1032/1029	OS	37	42.5/41.6	0.92 (0.80–1.06)	[8]
	**HORRAD**	PSA > 20ng/mL and bone lesions	RT to prostate + ADT	ADT	216/216	OS	47	45/43	0.90 (0.70–1.14)	[12]
**nmCRPC**	**ARAMIS**	PSA doubling time ≤ 10 months and basal PSA ≥ 2 ng/mL	Daro + ADT	Placebo + ADT	955/554	MFS	29	NE/NE	0.69 (0.53–0.88)	[13]
	**PROSPER**		Enza + ADT	Placebo + ADT	933/468	MFS	48	67/56.3	0.73 (0.61–0.89)	[14]
	**SPARTAN**		Apa + ADT	Placebo + ADT	806/401	MFS	52	73.9/59.9	0.78 (0.64–0.96)	[15]
	**TAX 327**	With or without symptoms	Doce + P	Mitoxantrone + P	335/337	OS	NA	19.2/16.3	0.79 (0.67–0.93)	[16]
**1st line mCRPC**	**COU-AA-302**	A/midly symptomatic pre-doce; no visceral mtx	AA + P + ADT	Placebo + P + ADT	546/542	rPFS, OS	49.2	34.7/30.3	0.81 (0.70–0.93)	[17]
	**PREVAIL**	A/midly symptomatic pre-doce	Enza + ADT	Placebo + ADT	872/845	rPFS, OS	69	36/31	0.83 (0.75–0.93)	[18]
	**IMPACT**	A/midly symptomatic pre-/post-doce; Gleason ≤ 7; no visceral mtx	Sipuleucel-T + ADT	Placebo + ADT	341/171	OS	34.1	25.8/21.7	0.78 (0.61–0.98)	[19]
	**IPAtential150**	A/midly symptomatic	AA + P + ipatasertib	AA + P + placebo	547/554	(bio)rPFS	19	NE/NE	NE	[20]
**≥2nd line mCRPC**	**COU-AA-301**	Post-doce	AA + P	Placebo + P	797/398	OS	20.2	15.8/11.2	0.74 (0.64–0.86)	[21]
	**TROPIC**	Post-doce	Cabazitaxel + P	Mitoxantrone + P	378/377	OS	25.5	NA/NA	0.72 (0.61–0.84)	[22]
	**AFFIRM**	Post-doce	Enza	Placebo	800/399	OS	14.4	18.4/13.6	0.63 (0.53–0.75)	[23]
	**ALSYMPCA**	Pre- and post-doce or unfit for doce; bone mtx and no visceral mtx	Radium-223	Placebo	614/307	OS	NA	14.9/11.3	0.70 (0.58–0.83)	[24]
	**CARD**	Post-doce and post-ARSi	Cabazitaxel	AA+P/Enza	129/126	IPFS	9.2	13.6/11	0.64 (0.46–0.89)	[25]
	**PROFOUND**	Post-ARSi and pre-/post-taxane	Olaparib	AA+P/Enza	162/83 *	(bio)IPFS	21	19.1/14.7*	0.69 (0.50–0.97) *	[26]
	**VISION**	Post-ARSi and 1–2 taxanes	LuPSMA	Standard of care	551/280	rPFS, OS	20.9	15.3/11.3	0.62 (0.52–0.74)	[27]

AA: abiraterone acetate; ADT: Androgen Deprivation Therapy; Apa: apalutamide; (bio): biomarker-defined population; ARSi: androgen-receptor signalling inhibitors; CI: Confidence Interval; Daro: darolutamide; Doce: docetaxel; Enza: enzalutamide; Exp: experimental; HR: Hazard Ratio; IPFS: image-guided progression-free survival; L/H: low/high; LuPSMA: Lutetium-177-PSMA-617; mCRPC: Metastatic Castration Resistant Prostate Cancer; MFS: Metastatic-free survival; mHSPC: Metastatic Hormonosensitive Prostate Cancer; mOS: median overall survival; mtx: metastases; NA: not availble; NE: Not Estimable; nmCRPC: Non-metastatic Castration Resistant Prostate Cancer; NSAA: nonsteroidal antiandrogen; P: prednisone; Ref; references; rPFS: Radiographic progression-free survival; RT: radiotherapy. * Results from BRCA1, BRCA2, ATM alterations Cohort.

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
