# Peer review of "Optimal Sequencing and Predictive Biomarkers in Patients with Advanced Prostate Cancer"

_cancers, 2021, doi:10.3390/cancers13184522_

Round 1

Reviewer 1 Report

The narrative review by Cattrini et al. aims to evaluate the available literature data to find the optimal sequencing approaches in patients with prostate cancer at different disease stages.

The first part of the review is of high clinical interest. In the last years, the therapeutic strategy for mCRPC radically changed. A number of new molecules have been registered for mCRPC, and several emerging compounds are currently in the course of their validation. In this scenario, no standardized approach to correctly sequence the available treatment options is still defined. For these reasons, I think that the readers of Cancers will highly appreciate this content.

However, while paragraph 2 is well written, comprehensive, and reasonably organized, I think that paragraph 3 should be implemented. The authors extensively analyzed the molecular predictive biomarkers able to identify biomarker-selected populations having higher chances to benefit from specific treatments. However, an increasing number of clinical, biochemical, and imaging data have been proposed as potential tools to improve the treatment selection process at a single patient level. An emblematic example is represented by molecular imaging with Positron Emission Tomography, which may non-invasively describe many characteristics of the prostate cancer biology (i.e., PSMA expression, bone osteoblastic activity, choline or glucose metabolism) and the heterogeneity of these biologic processes between metastatic sites, thus potentially guiding the systemic treatment selection.

Author Response

We thank the reviewer for his/her comment. We agree that several advances have been achieved with molecular imaging. In this regard, a specific paragraph is dedicated to the role of PET-PSMA and Lutetium. However, to our knowledge, there is no evidence that specific imaging findings may be related to response to therapy. Of note, all phase 3 trials in both metastatic hormone-sensitive and castration resistant patients have used standard imaging for the staging of patients and there is no current evidence that staging with more advanced techniques could improve the outcome of patients and predict response to therapies. If the reviewer is able to provide adequate bibliography, we will implement the paper with specific data that could help treatment selection.

Reviewer 2 Report

an interesting manuscript showing the correlation of predictive molacular genetic markers and treatment  for patients with prostate cancer. I suggest to  add a paragraph concerning biochemical markers (i.e. TK, TK210, mindin) which if taken together molecular genetic biomarkers mentioned in the manuscript  significantly improve diagnostic specificity and may help to focus treatment. 

Author Response

We thank the reviewer for his/her comment. We have not found scientific literature that supports the role of TK, TK210 or mindin in predicting response to therapies in prostate cancer, potentially affecting treatment sequence. If the reviewer is able to provide adequate bibliography, we will implement the paper with specific data on biochemical biomarkers that could help treatment selection.

Reviewer 3 Report

The treatment landscape of advanced prostate cancer has completely changed during the last decades. In this publication, Cattrini and al. explore optimal therapy sequencing and predictive biomarkers that could be used in patients with advanced prostate cancer.

General comment:

The manuscript is well-written, well-structured, logic and comprehensive.

Minor points:

The authors could briefly mention Denosumab (a human monoclonal antibody to RANKL that significantly suppresses bone resorption).

For platinum-based chemotherapy, the higher rates of efficacy in the presence of DNA damage repair defects could be mentioned.

Page 1, line 106: To be more precise, the authors could write ''docetaxel added to ADT was found to be superior to ADT alone''.

Page 11, line 556: The authors did not explain the ''ITT'' abbreviation.

English language should be corrected in the following lines: Page 3, lines 178-179; page 8, line 423; page 12, line 613-614.

Author Response

We thank the reviewer for his/her comments. We have addressed all points suggested by the reviewer (modifications in red in the text). Specifically, a paragraph related to the role of bone-targeted agents has been added, and another paragraph describes the effect of platinum-based therapy in DDR-deficient prostate cancer. English editing has been performed.

Round 2

Reviewer 1 Report

I acknowledge the answer of the authors. However, I disagree for the following reasons:   

1) The VISION trial is a phase III study in which enrolled patients should have had PSMA-positive disease on the basis of a central review of 68Ga-PSMA-11 staging scans. As elegantly debated in this recent editorial (PMID: 34446452), in the era of theranostics, it is highly questionable the use of 177Lu-PSMA therapy on the sole basis of standard imaging. 

2) The TheraP phase II study required a baseline 68Ga-PSMA11 PET, as well as FDG PET to exclude patients with metabolically active (perhaps low-differentiated) disease sites lacking PSMA expression. The outcomes, not surprisingly, appeared superior to those reported in VISION and might serve as a further argument for further optimizing the eligibility screening. 

3) Many proof-of-concept retrospective, prospective, phase I-II studies have been also published regarding the use of PET imaging with different radiotracers (mainly PSMA, choline, FDHT, FDG) for the systemic treatment response prediction in CRPC. The authors might read this review PMID: 33809749.  

Author Response

We thank the reviewer for her/his comments and for having provided appropriate bibliography. We have implemented the Lu-PSMA paragraph and we have added an additional paragraph in the biomarker section (all changes are highlighted in red).